# Cortical Thickness Links Impulsive Personality Traits and Risky Behavior

**DOI:** 10.3390/brainsci9120373

**Published:** 2019-12-13

**Authors:** Rickie Miglin, Nadia Bounoua, Shelly Goodling, Ana Sheehan, Jeffrey M. Spielberg, Naomi Sadeh

**Affiliations:** 1Department of Psychological & Brain Sciences, University of Delaware, Newark, DE 19713, USA; nbounoua@udel.edu (N.B.); ansheeha@udel.edu (A.S.); jmsp@udel.edu (J.M.S.); 2Department of Psychology, York College of Pennsylvania, York, PA 17403, USA; sgoodling@ycp.edu

**Keywords:** UPPS-P, cortical thickness, urgency, sensation seeking

## Abstract

Impulsive personality traits are often predictive of risky behavior, but not much is known about the neurobiological basis of this relationship. We investigated whether thickness of the cortical mantle varied as a function of impulsive traits and whether such variation also explained recent risky behavior. A community sample of 107 adults (ages 18–55; 54.2% men) completed self-report measures of impulsive traits and risky behavior followed by a neuroimaging protocol. Using the three-factor model of impulsive traits derived from the UPPS-P Impulsive Behavior Scale, analysis of the entire cortical mantle identified three thickness clusters that related to impulsive traits. Sensation seeking was negatively related to thickness in the right pericalcarine cortex, whereas impulsive urgency was positively associated with thickness in the left superior parietal and right paracentral lobule. Notably, follow-up analyses showed that thickness in the right pericalcarine cortex also related to recent risky behavior, with the identified cluster mediating the association between sensation seeking and risky behavior. Findings suggest that reduced thickness in the pericalcarine region partially explains the link between sensation seeking and the tendency to engage in risky behavior, providing new insight into the neurobiological basis of these relationships.

## 1. Introduction

Although a considerable amount of research has investigated the relationship between impulsive personality traits and engagement in risky behavior, much less is known about the neurobiological substrates associated with these personality characteristics. Identifying the neuroanatomical structures associated with dispositions to engage in risky behavior can reveal novel neural mechanisms and psychological processes that serve to initiate and maintain these harmful behaviors. Furthermore, this type of examination can be useful for identifying diverse neurobiological pathways to impulsive and risky behavior, given that etiological heterogeneity may not be apparent at the phenotypic level. The goals of this study are (a) to examine the neuroanatomical correlates of personality traits linked to risky behavior, and (b) determine the relevance of these neural markers for explaining recent engagement in risky behavior. 

The personality traits that predispose some individuals to act more rashly than others are known to be complex and multidimensional [1,2]. One prominent model of impulsive traits that was developed based on the Five Factor Model of Personality [3] identifies three higher-order factors that dispose individuals to rash action, specifically low conscientiousness, sensation seeking, and impulsive urgency. Low conscientiousness is characterized by rash action that results from a lack of persistence and planning, sensation seeking is impulsive behavior that is marked by the pursuit of exciting or thrilling experiences, and impulsive urgency captures the tendency to act impulsively when experiencing strong emotions [4,5,6].

Although all three factors tap impulsive tendencies, they are theorized to represent separable, albeit overlapping, pathways to impulsive action, potentially with distinct etiologies [7]. It has been posited that the impulsive urgency factor is related to negative and positive reinforcement sensitivity, and that these represent mechanisms driving the connection between urgency and risky behaviors [4]. Sensation seeking, though similarly associated with high reward sensitivity and low punishment sensitivity [8], may also be driven by a high threshold for fear [9], low pain sensitivity [10], and/or increased dopamine in response to stress [11,12]. The low conscientiousness factor has been linked with low levels of executive functioning [13], low attentional control [14], and possibly insufficient sensitivity to reinforcement contingencies [15]. Thus, there is some evidence that impulsive personality traits are associated with related, but distinct, etiologic pathways or mechanisms.

### 1.1. Neuroanatomical Correlates of Impulsivity

Studies on the neurobiological correlates of impulsive traits have the potential to uncover other underlying processes that may contribute to impulsive and risky phenotypes. Extant neuroimaging research has demonstrated associations between brain structure and impulsivity in adults [16,17,18]. For example, studies have shown positive correlations between self-reported impulsivity and volume of the anterior cingulate gyrus and dorsolateral prefrontal cortex [19]. More recent work has sought to extend this literature by examining neural correlates of emotion-related impulsivity. For example, after controlling for other impulsivity-based personality traits and neuroticism, negative urgency has been associated with reduced volume in the dorsomedial prefrontal cortex and left ventral striatum, regions associated with emotion-based decision-making [20]. Similarly, another study found that sensation seeking (indexed with the novelty seeking scale from the Temperament and Character Inventory) correlated with reduced cortical thickness in neural regions associated with cognitive control including the anterior cingulate and middle frontal gyrus [21]. Although these studies provide preliminary support for neurobiological correlates of emotion-based impulsivity, this area of research is relatively limited [22], with no studies to date examining the structural correlates of the three-factor model of impulsive traits. 

### 1.2. Impulsive Traits and Risky Behavior

Impulsive personality traits are established predictors of engagement in a range of risky behaviors. Impulsive urgency has emerged as a predictor of alcohol and substance misuse [5,23], suicidality, and non-suicidal self-injury (NSSI) [24,25], and disordered eating [26]. Similarly, sensation seeking is a robust correlate of problematic alcohol and substance use [23,27,28], and at times, suicidality and NSSI [29,30]. Importantly, sensation seeking has been shown to correlate strongly with the frequency of engagement in certain kinds of risky behavior, whereas impulsive urgency has been found to correlate with problematic levels of risky behavior [31]. Individuals also low in conscientiousness are more likely to engage in substance use [32] and suicidality [33] as well as risky driving [34] and risky sexual behavior [35].

In spite of the evidence that impulsive traits contribute to a spectrum of risky behaviors, most research has focused on the connection between these traits and specific forms of risky behavior such as substance use and self-harm. There has been relatively less emphasis on examining how impulsive traits relate to engagement in risky and self-destructive behavior more broadly. This is a notable gap in previous research given extant evidence that diverse behavioral manifestations of risky behaviors often co-occur within individuals across the lifespan and reflect latent traits [36,37].

### 1.3. Current Study

Past research has demonstrated a robust link between impulsive personality traits and risky behavior, but more work is needed to identify the neural characteristics that instantiate these relationships as well as possible understudied associations with broad engagement in risky behavior. The first aim of the study was to examine whether the three-factor model of impulsive traits shows reliable associations with cortical thickness, a measure of grey matter density in the cerebral cortex. The second and third aims were to investigate whether such variations in cortical thickness (i) also relate to engagement in risky behavior and (ii) explain the relationship between impulsive traits and risky behaviors. We hypothesized that cortical thickness would vary uniquely as a function of the three-factor model of impulsive traits specified by the UPPS-P (sensation seeking, impulsive urgency, and low conscientiousness) and that those differences in brain structure may partially explain the association between impulsive traits and engagement in risky behavior. Although we examined cortical thickness as an intermediate mechanism of the relationship between trait impulsivity and engagement in risky behavior, this type of causal model cannot be rigorously tested using cross-sectional data, because the direction of the effects cannot be determined. Therefore, we also tested an alternative causal model.

## 2. Materials and Methods

### 2.1. Participants

A total of 115 adults completed a battery of questionnaires and a neuroimaging protocol. Eight participants were excluded due to incidental magnetic resonance imaging (MRI) findings or excessive motion (all images visually inspected for quality), resulting in a final sample of 107 adults (M/SD age = 32.1/9.4, 45.8% female). Participants were recruited from the community through the use of fliers and online postings. The sample was socioeconomically and racially diverse, as described in Table 1. The average household income in the sample was less than $43,000 for the last year, and the majority of participants came from communities with high rates of violent and non-violent crime (https://www.neighborhoodscout.com/de/wilmington/crime on 12/2/19) [38]. More than half of the sample reported being arrested at least once in their lifetime (see Table 1). Taken together, these characteristics suggest that the current sample may be at relatively higher risk for engagement in risky behavior than other samples. Participants identified as Caucasian (49.5%), Black or African American (39.3%), Asian American (6.5%), or “Other” (4.7%), and 16.8% identified as Hispanic or Latino. 

Interested individuals were recruited to the study if they were between the ages of 18–55 and fluent in the English language. Participants were excluded if they demonstrated evidence of psychosis, had an estimated IQ in the Intellectual Disability range, had serious medical or neurological conditions (e.g., epilepsy), a history of three or more head injuries with loss of consciousness, and/or other MRI contraindications (e.g., metal implants, current pregnancy). Written and oral consent was obtained from all individuals prior to participation in the study. All procedures were approved by the university’s Institutional Review Board (Protocol Nos.: 1073423-17 and 1361164-8). 

### 2.2. Measures

Impulsive Traits. The UPPS-P Impulsive Behavior Scale (UPPS-P) [39] is a 59-item self-reporting measure that assesses distinct personality traits that lead to engagement in impulsive behavior: negative urgency, premeditation, perseverance, sensation seeking, and positive urgency. Items are scored on a scale from 1 (Agree Strongly) to 4 (Disagree Strongly). Based on research indicating the five UPPS-P dimensions can be modeled using a higher-order three-factor solution [6], analyses focused on this factor structure. An impulsive urgency factor was created to reflect the tendency to engage in impulsive behaviors when experiencing strong emotions by averaging the negative urgency subscale (12 items; e.g., “When I am upset, I often act without thinking.”) and the positive urgency subscale (14 items; e.g., “I tend to lose control when I am in a great mood.”). The tendency to act rashly during the pursuit of exciting or thrilling experiences was assessed with the sensation seeking subscale (12 items; “I generally seek new and exciting experiences and sensations.”). A low conscientiousness factor was created to reflect the tendency to engage in impulsive behavior as a function of lack of planning and persistence by averaging the premeditation (11 items; e.g., “My thinking is usually careful and purposeful.”) and perseverance (10 items; e.g., “I finish what I start.”) subscales. All three factors showed good to excellent reliability in our sample (Cronbach’s alphas ranged from 0.85 to 0.94). 

Risky Behaviors. The Risky, Impulsive, and Self-Destructive Behavior Questionnaire (RISQ) [36] was used to assess the frequency of past month and lifetime risk-taking behaviors. The RISQ is a 38-item self-reporting questionnaire that measures the frequency of a range of risky and impulsive behaviors including drug use, aggression, self-harming behaviors, gambling, risky sexual behavior, heavy alcohol use, impulsive eating, and reckless driving/spending behavior. Participants reported how many times they engaged in the past month and throughout their lifetime. As described elsewhere [36], responses were categorized into five bins that constrained the range of possible responses at the high end of the distribution: 0, 1–10, 11–50, 51–100, >100 times. Positive skewness was further reduced using a Blom transformation. 

MRI Data Acquisition. Data were collected at the University of Delaware using a Siemens 3T Magnetom Prisma scanner with a 64-channel head coil. T1-weighted multi-echo MPRAGE anatomical scan (resolution = 1 mm^3^, TR = 2530 ms, TEs = 1.69, 3.55, 5.41, 7.27 ms) was collected, which has the advantage of less distortion and higher contrast than standard MPRAGE sequences, resulting in more reliable cortical models [40]. A T2-weighted variable flip-angle turbo spin-echo scan (resolution = 1 mm^3^, TR = 3200 ms, TE = 564 ms) was collected, which is used in FreeSurfer to better differentiate the gray-matter-dura boundary.

### 2.3. Data Analysis

Cortical Thickness. The thickness of the cortical mantle at each vertex was estimated using FreeSurfer’s [41] (v6) standard morphometric pipeline [42,43]. FreeSurfer’s method for constructing and transforming the cerebral cortex involves a high resolution, surface-based averaging technique that aligns cortical folding patterns. The technical details of these procedures are described in prior publications [44,45]. Cortical thickness was calculated as the closest distance from the grey/white boundary to the grey/cerebrospinal fluid boundary at each vertex on the tessellated surface [46]. The data were spatially smoothed using a Gaussian kernel of 10 mm full width at half maximum (FWHM) and a nearest-neighbor averaging algorithm where each vertex’s value is averaged with those of its neighbors. Spatial smoothing ensures that the data more closely approximate a continuous field of random values, which is a necessary assumption of the Monte Carlo simulations later used for multiple comparisons correction [47]. 

As no prior research has examined associations between the three-factor model of impulsivity and cortical thickness, we probed the entire cortical mantle using an exploratory vertex-wise approach to assess the association between impulsive personality traits and cortical thickness. General linear models were conducted separately for each hemisphere using FreeSurfer’s built-in QDEC application. Impulsive urgency, sensation seeking, and low conscientiousness were entered as the explanatory variable in separate analyses, with age, biological sex, and body mass index (BMI) entered as covariates of no interest. The vertex-wise significance threshold was set at *p* < 0.01. Cluster-wise correction for multiple comparisons was computed using a Monte Carlo null-Z simulation with 10,000 iterations and a cluster-wise threshold correction of 1.3 (*p* < 0.05) (methods based on Hagler et al., 2006 [47]). The Monte Carlo null-Z simulation used by FreeSurfer’s QDEC is based on an iterative process that uses random image generation, individual voxel probability thresholding, and minimum cluster size thresholding. The frequency count of cluster sizes determines the probability of a false positive detection per image [47]. This methodology has been validated for use with cortical thickness data [48].

Regression and Mediation Analyses. Mean thickness was extracted for the clusters associated with impulsive traits in the above analyses. We ran hierarchical linear regressions in SPSS v26 to examine whether mean thickness in any of the clusters that emerged was related to past month or lifetime engagement in risky behavior. For any of the clusters showing significant associations with recent risky behavior, we ran mediation models in Mplus v8.0 [49] to determine whether thickness mediated the relationship between impulsive traits and risky behavior. The mediation analyses focused on past month (rather than lifetime) risky behavior based on the theorized temporal ordering of the predictor (impulsive personality traits), mediator (cortical thickness), and outcome variables (recent risky behavior). All of the variables, except biological sex (men = 1; women = 0), were continuous and met the distributional assumptions for the mediation and regression models. More specifically, the self-report and neuroimaging variables were normally distributed and did not evidence excessive skewness or kurtosis (all values were between 1.0 and −1.0). For the mediation analyses, we used maximum likelihood estimation with robust standard errors (MLR) in Mplus using the “model indirect” procedure. All tests were two-tailed and included age, biological sex, and BMI as covariates. Two participants were missing RISQ self-report data due to unreliable response patterns.

## 3. Results

### 3.1. Impulsive Traits and Risky Behavior

Participants reported a range of risky behaviors in the past month (RISQ Total: M/SD = 5.52/5.05, Min/Max = 0.00/25.00), and the vast majority of the sample endorsed at least one instance of risky behavior in the 30 days prior to the assessment (88.6%). The most prevalent types of risky behavior were reckless driving/spending behaviors (66.7%) and illicit drug use (46.6%), followed by dysregulated eating (36.1%), alcohol misuse (24.7%), gambling (25.7%), risky sexual behavior (15.2%), aggression (12.4%), and self-harm behaviors (5.8%). 

As described in Table 2, past month risky behavior was positively associated with trait sensation seeking and impulsive urgency, but not with low conscientiousness. Lifetime frequency of risky behavior, however, correlated positively with all three factors. 

### 3.2. Impulsive Trait Associations with Cortical Thickness

Vertex-wise analysis of the cortical mantle revealed multiple regions in which thickness was associated with impulsive traits and cortical thickness and survived correction for multiple comparisons (Figure 1, Table 3). Sensation seeking correlated negatively with thickness in a cluster that peaked in the right pericalcarine cortex and also spanned the cuneus, occipital pole, middle occipital cortex, and superior occipital gyrus. Impulsive urgency—the tendency to act rashly when experiencing strong emotions—was related to thickness in two clusters (Figure 2, Table 3): the peak of the first cluster was located in the left precuneus and also spanned the superior parietal lobule (SPL), superior occipital gyrus, and cuneus; the second cluster peaked in right paracentral lobule, also spanning the precuneus and posterior cingulate cortex. No clusters emerged in which low conscientiousness was related to cortical thickness.

### 3.3. Cortical Thickness Associations with Risky Behavior

We extracted mean thickness from each of the clusters identified above and entered these as simultaneous predictors into hierarchical linear regressions, with either recent or lifetime risky behavior as the dependent variable, and age and sex entered as covariates of no interest (Table 4). Lower thickness in the right pericalcarine cluster (linked with sensation seeking) was related to more risky behavior in both the past month and over the lifespan. Thickness in left SPL cluster (linked with urgency) was positively related to lifetime (but not recent) risky behavior, as was thickness in the right paracentral cluster at a trend-level.

### 3.4. Cortical Thickness Mediates the Association between Sensation Seeking and Recent Risky Behavior 

Finally, we tested two models that simultaneously assessed relations between trait impulsivity, cortical thickness, and recent risky behavior. First, we examined whether thickness in the right pericalcarine cluster (the only cluster related to risky behavior) acted as a mediator linking sensation seeking to past month risky behavior. The direct effects of sensation seeking on right pericalcarine thickness (β = −0.41, *p* < 0.001) and right pericalcarine thickness on past month risky behavior (β = −0.24, *p* = 0.028) were both significant. The direct path from sensation seeking to risky behavior became non-significant when pericalcarine thickness was entered in the model (β = 0.18, *p* = 0.11). Importantly, the indirect effect of sensation seeking on risky behavior was significant (β = 0.10, *p* = 0.034), indicating that the right pericalcarine thickness mediated the relationship between sensation seeking and past month risky behavior. In total, the mediation model explained 19.7% of the variance in cortical thickness in the right pericalcarine cluster (*p* = 0.002), and 20.1% of the variance in the past month risky behavior (*p* = 0.001).

Second, we examined an alternative model whereby trait sensation seeking mediated the association between thickness in the right pericalcarine cluster and recent risky behavior. The indirect effect from cortical thickness to risky behavior via sensation seeking was not significant (β = −0.06, *p* = 0.15), which is likely due to the fact that the direct effect from the cortical thickness in that cluster to risky behavior remained significant with sensation seeking in the model (β = −0.24, *p* = 0.03). This alternative model explained 14.5% of the variance in recent risky behavior (*p* = 0.001). 

## 4. Discussion

A large body of research has demonstrated that impulsivity contributes to risky behavior, but far less is known about the neuroanatomical substrates that mediate this association. To further understand the neurobiological links between impulsive traits and risky behavior, we examined whether (i) cortical thickness varied as a function of impulsive personality traits, (ii) impulsivity-related variation in cortical thickness also covaried with recent risky behavior, and (iii) whether cortical thickness mediated the relationship between impulsivity and risky behavior. In a sample of community adults who reported a range of risky behaviors, the three-factor model of impulsive traits derived from the UPPS-P showed distinct associations with cortical thickness in several brain regions. Three clusters survived correction for multiple comparisons: higher trait sensation seeking was associated with less cortical thickness in the right pericalcarine cortex, whereas higher trait urgency was linked to greater cortical thickness in the left superior parietal lobule (SPL) and right paracentral lobule. Notably, mean thickness extracted from the pericalcarine and SPL clusters were associated with lifetime risky behavior, and the pericalcarine cluster was also related to recent risky behavior. Finally, we found evidence that thickness in the pericalcarine cluster mediated the relationship between sensation seeking and recent risky behavior. Together, these results provide new insights into potential neuroanatomical mechanisms instantiating impulsive, risky behavior, particularly a potentially novel intermediate mechanism linking trait sensation seeking with risky behavior.

Despite widespread use of the UPPS-P to index personality characteristics that predispose individuals to rash action, there is a scarcity of research on the neuroanatomical correlates of the UPPS-P three-factor model of impulsive personality traits. However, the methodological approach used herein is promising because it has the potential to yield information about distinct neurobiological pathways to rash action and risky behavior. This etiological heterogeneity may not be apparent when only examining behavioral phenotypes of impulsivity. Given the lack of previous research, we conducted an exploratory analysis that examined the entire cortical mantle for regions where thickness was related to each impulsive personality dimension. Unique patterns of associations were observed for the sensation seeking and impulsive urgency dimensions, although no regions emerged that were related to low conscientiousness. 

Sensation seeking predicted lower thickness in a cluster centered on the pericalcarine cortex that also spanned the occipital pole, cuneus, and middle and superior occipital gyrus. The pericalcarine and posterior occipital cortices make up part of the primary visual cortex V1 [50]. Surface area in V1 is known to vary significantly between individuals and is thought to contribute substantially to variability in conscious experience [51]. Previous research has shown increased neural activity in a similar region when individuals high on sensation seeking are presented novel versus familiar objects [52]. Additionally, Chase et al. [53] found a positive relationship between sensation seeking and reward expectancy activity in the occipital cortex. Other research has also found links between sensation seeking and lower cortical thickness in the pericalcarine cortex, along with lower middle frontal gyrus and supramarginal gyrus volumes [21]. Thus, the present findings augment previous research by replicating associations between sensation seeking and the pericalcarine cortex region via the widely-used UPPS-P.

In contrast to rash action that is motivated by sensation seeking, impulsive urgency predicted greater cortical thickness in two clusters. The first cluster was located in the left superior parietal lobule, peaking in the precuneus and spanning the superior occipital gyrus, and cuneus. The SPL is known to play an important role in cognitive, perceptive, and motor-related process [54] including visual information processing and spatial orientation [55]. Additionally, the precuneus (part of the SPL) is thought to be central to the default mode network, and crucial for self-referential processing and regulating attentional states [56,57]. Urgency also predicted greater thickness in the right precuneus, spanning the paracentral lobule, part of the supplementary motor area (SMA) [58], and posterior cingulate gyrus, thought to play a role supporting internally-directed cognition [59]. The present finding of greater thickness in these regions extends research that has linked negative urgency to reduced grey matter volume in the dorsomedial frontal, temporal pole, and ventral striatal volumes in healthy adults [20]. Given the functions associated with the identified brain regions, the positive association between urgency and thickness in these clusters may reflect a sensitivity to emotional salience or a compensatory mechanism for regulating responses to affective stimuli and related impulsive urges in individuals high on this dimension of impulsivity. These possibilities are speculative, but could be addressed in future research with functional neuroimaging tasks. 

The urgency-related brain regions of the current study may differ from this previous work for several reasons. The aforementioned study examined brain morphology using a five-factor rather than a three-factor model of the UPPS-P, which collapses both forms of impulsive urgency (negative and positive) on a single affective impulsivity arousal factor. Additionally, we recruited an unselected community sample rather than a sample of healthy adults, and the high rates of risky behavior reported by our sample suggests they may be at elevated-risk for impulse control and related problems. Finally, the study reported above [20] used voxel-based-morphometry (VBM) rather than cortical thickness analyses to examine brain structure, and VBM is known to be more closely related to surface area than cortical thickness [60].

Consistent with previous research, higher levels of sensation seeking and impulsive urgency were related to greater engagement in recent risky behavior. Additionally, thickness in the clusters that were associated with impulsive traits were also associated with lifetime engagement in risky behavior. However, only diminished thickness in the sensation seeking pericalcarine cluster corresponded to higher engagement in risky behaviors in the past month. Follow-up analysis determined that thickness in the right pericalcarine cluster significantly mediated the relationship between sensation seeking and past month risky behavior. This finding suggests that reduced thickness in this cluster plays an important role in maintaining the relationship between sensation seeking and risky behavior, though whether sensation seeking precedes brain structure, or vice versa, is unclear. Future research should test these associations using a longitudinal design to tease apart the temporal ordering of impulsive personality traits and variations in brain structure. 

It is notable that sensation seeking and impulsive urgency were associated with variations in cortical thickness in clusters that spanned the parietal and occipital lobes and evidenced opposing relations with thickness. These findings provide new evidence that the impulsive personality traits reflect distinct etiological pathways to rash behavior. Despite the differences that were observed for the impulsive traits in terms of their associated brain regions and the direction of the association with thickness, the clusters associated with sensation seeking and impulsive urgency were both related to lifetime engagement in risky behavior, further validating their construct validity. 

Although the lack of an association between low conscientiousness and thickness was unexpected, it may be that the sample was underpowered to detect an effect. Other studies have found significant differences in gray matter volumes related to the low conscientiousness factor between drug using individuals and healthy controls, which may partially explain why we were unable to detect effects in an unselected community sample [61]. Additionally, urgency (particularly negative urgency), has been shown across multiple populations to be the most robust prospective predictor of risky and self-destructive behavior (e.g., binge-drinking and drug use) of the UPPS-P domains [62,63]. It is possible that since the urgency factor most strongly predisposes individuals to engage in behaviors that are known to influence brain morphology (such as substance use), this enabled us to more easily observe effects related to this factor compared to the low conscientiousness factor. However, the lack of association between conscientiousness and cortical thickness is surprising and may suggest the need for a larger sample size or sample with different characteristics (e.g., a less socially disadvantaged sample). 

An important extension of these findings will be to examine the functional implications of the observed structural differences associated with the three-factor model of impulsive traits. In particular, examining whether the regions that showed variation in thickness as a function of impulsive traits also show alterations in network functional connectivity would help elucidate the neural processes affected by these structural abnormalities. Additionally, future studies should examine whether thickness in these clusters predicts future risky behavior in multiple ways, for example, by measuring thickness in these regions and then either administering a behavioral risk-taking task or measuring participants’ engagement in risky behavior weeks or months after completing the MRI protocol. This would strengthen the argument that thickness in the identified clusters causes elevated levels of risk-taking.

As with all research, the present results must be interpreted in the context of the study limitations. First, impulsivity, cortical thickness, and past-month risky behaviors were assessed cross-sectionally, which limits our ability to make strong inferences about the causal relationships between these variables. However, our models are consistent with theoretical models that suggest that trait-level personality factors are relatively stable across the lifespan and, thus, drive engagement in risky behaviors [64,65]. Furthermore, cortical thickness is a relatively stable, and therefore trait-like, neurobiological indicator, which increases the likelihood that it predicts past month risky behavior (rather than recent behavior impacting thickness). Nonetheless, studies using prospective designs would only enhance understanding of the relationships between impulsive traits, cortical thickness, and risky behavior, and replication studies need to be conducted to confirm our findings. Second, impulsivity and risk-taking behaviors were assessed using self-report measures, which introduces the possibility for bias and error in the measurements of these constructs. Although, it seems unlikely that response bias in the self-report measures would impact the cortical thickness results, it does introduce error and may have masked true associations. At the same time, our measure of trait impulsivity is one of the most well-validated assays of the impulsivity construct and is known to correlate reliably with behavioral tasks of impulsivity [66], and shows test–retest reliability [67]. Future research should aim to replicate these findings using additional objective measures of impulsivity such as laboratory tasks (e.g., delay discounting or stop-signal tasks). Third, we only examined relations with cortical thickness and it is likely that other neural markers are also differentially related to the three-factor model of impulsive traits such as subcortical volumes, a possibility that should be explored to fully map the neuroanatomical markers of these factors that predispose to rash action. Finally, the findings suggest that the current sample reported rates of risky behavior that are higher than those typically observed in college students and community adults [36]. Although the range and frequency of behaviors reported by participants in this study is a strength for identifying the neural correlates of risky behavior, the results may not be generalizable to lower-risk populations.

The current study also has several notable strengths. First, the sample was an ethnically and socioeconomically diverse group of community adults that reported high rates of engagement in multiple types of risky behavior. While most studies tend to focus on specific types of risky behavior, this study took a broader approach by measuring the frequency of risky behavior across multiple domains (e.g., substance use, gambling, reckless driving, self-harm) to examine the overall severity of an individual’s engagement in risky behavior, which better captures the frequent co-occurrence of risky phenotypes. This study was also one of the first to explore the three-factor model of impulsive traits derived from the widely-used UPPS-P scale. Given the large number of studies that have used the UPPS-P to study impulsive personality traits, the current findings expand the nomological network of these constructs. Finally, the sample recruited for this study was larger than most other studies that have examined impulsive personality traits and cortical thickness in at-risk (e.g., non-college student) populations. 

In summary, the results provide initial data on the neuroanatomical correlates of the three-factor model of impulsive traits, with findings suggesting that sensation seeking and impulsive urgency evidence distinct relationships with thickness across the cerebral cortex. Of note, one thickness cluster related to impulsive traits was also predictive of recent engagement in risky behavior, and mediation analysis suggested reduced thickness in the pericalcarine cortex partially explained the link between sensation seeking and the tendency to engage in risky behavior. Together, these results contribute new insights into the neurobiological basis of the relationship between impulsive traits and risky behavior in adulthood. 

## Figures and Tables

**Figure 1 brainsci-09-00373-f001:**
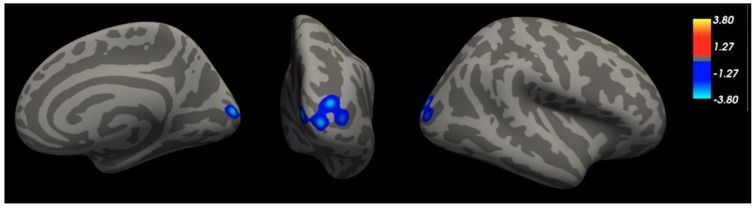
Sensation seeking relates to lower cortical thickness in a cluster of the right hemisphere, adjusting for age, sex, and BMI. Cluster spanned pericalcarine cortex, cuneus, occipital pole, superior occipital gyrus, middle occipital gyrus.

**Figure 2 brainsci-09-00373-f002:**
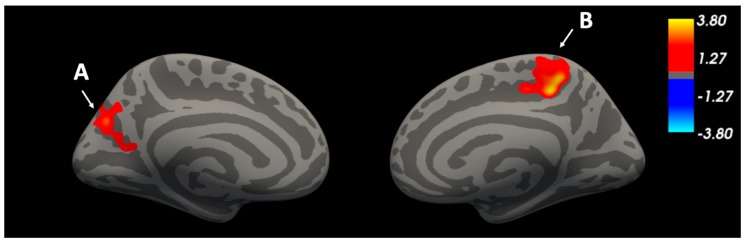
Impulsive urgency predicts greater cortical thickness in two clusters, adjusting for age, sex, and BMI. (**A**) Left superior parietal lobule, precuneus, superior occipital gyrus, and cuneus. (**B**) Right paracentral lobule, precuneus, and posterior cingulate gyrus.

**Table 1 brainsci-09-00373-t001:** Sample characteristics and study variables (*N* = 107).

**Demographics**	
Age (M/SD)	32.1/9.4
Biological Sex (*n*, %)	
Female	49/ 45.8%
Male	58/ 54.2%
BMI (M/SD)	27.0/5.2
Ethnicity (*n*, %)	
White	53/49.5%
Black/African American	42/39.3%
Hispanic/Latino	18/16.8%
Asian	7/6.5%
Other	5/4.7%
Household Income (M/SD)	$42,629/$38,301
Justice System (*n*, %)	58/54.2%
**Study Variables**	
Impulsive Urgency (M/SD)	2.2/0.7
Sensation Seeking (M/SD)	2.8/0.7
Low Conscientiousness (M/SD)	1.8/0.4
Past Month Risky Behaviors (M/SD)	5.5/5.1
Lifetime Risky Behaviors (M/SD)	30.2/19.4

**Table 2 brainsci-09-00373-t002:** Bivariate correlations among impulsive traits, risky behavior, and covariates.

	1	2	3	4	5	6	7
1. Sensation Seeking	--						
2. Impulsive Urgency	0.25 *	--					
3. Low Conscientiousness	−0.01	0.50 **	--				
4. Risky Behavior (Past Month)	0.30 **	0.46 **	0.16	--			
5. Risky Behavior (Lifetime)	0.21 *	0.55 **	0.32 **	0.71 **	--		
6. Age	−0.24 *	−0.04	−0.10	0.04	0.23 *	--	
7. Male Sex	0.37 **	0.13	0.22 *	0.17	0.17	0.06	--
8. BMI	−0.06	−0.06	−0.01	0.05	0.01	0.24 *	0.01

Note. * *p* < 0.05, ** *p* < 0.01. Values represent Pearson correlation coefficients except for associations with Male Sex, which are Spearman correlation coefficients.

**Table 3 brainsci-09-00373-t003:** Regions of significant correlation between UPPS-P traits and cortical thickness.

Hemisphere	Annotation	Peak *F*-Value	Peak MNI (x,y,z)	No. of Vertices	Cluster Size (mm^2^)
Sensation Seeking					
RH	Pericalcarine CUN OP SOG MOG	−3.50	5.9, −87.8, 7.5	1707	1262.62
Impulsive Urgency					
LH	PCUN SPG SOG CUN	2.71	−19.0, −72.8, 26.2	1783	969.34
RH	Paracentral PCUN PCG	3.81	17.8, −40.2, 42.4	2395	833.68

Note. *N* = 107. All clusters survived Monte Carlo Simulation correction for multiple comparisons (*p* < 0.05). Covariates included age, sex, and BMI. RH = right hemisphere. LH = left hemisphere. CUN = cuneus. OP = occipital pole. SOG = superior occipital gyrus. MOG = middle occipital gyrus. PCUN = precuneus. SPG = superior parietal gyrus. PCG = posterior cingulate gyrus.

**Table 4 brainsci-09-00373-t004:** Frequency of risky behavior regressed on thickness clusters associated with impulsive personality traits.

	Total Past Month Risky Behavior	Total Lifetime Risky Behavior
	β (SE)	β (SE)
**Step 1**		
Age	0.03 (0.05)	0.22 (0.20)
Sex	0.11 (1.00)	0.11 (3.74)
**Step 2**		
RH Pericalcarine (Sensation Seeking)	−0.34 (3.59) **	−0.32 (13.00) **
RH Paracentral (Impulsive Urgency)	0.20 (2.64)	0.22 (9.58) ^+^
LH Superior Parietal (Impulsive Urgency)	0.20 (3.78)	0.34 (13.70) **

Note. RH = right hemisphere. LH = left hemisphere. The impulsive trait that was associated with thickness in the cluster is identified in parentheses. Total Past Month Risky Behavior: Step 1 R^2^ = 0.01. Step 2 ΔR^2^ = 0.11 **. Total Lifetime Month Risky Behavior: Step 1 R^2^ = 0.07 *. Step 2 ΔR^2^ = 0.16 **. ^+^
*p* < 0.06, * *p* < 0.05, ** *p* < 0.01.

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
