# Peer review of "Cortical Thickness Links Impulsive Personality Traits and Risky Behavior"

_brainsci, 2019, doi:10.3390/brainsci9120373_

Round 1

Reviewer 1 Report

The paper concerns an important question of the neural basis of impulsive personality traits and risky behavior. The present state of psychological and neuroimaging research of these phenomena is adequately reviewed in the Introduction. The study involved a substantial sample of healthy volunteers, with questionnaire-based assessment and morphometric MRI acquisition performed according to the contemporary standards. The simultaneous evaluation of impulsivity, risky behavior, and cortical thickness for probing their relationships has the potential to expand the understanding of causal links between these aspects of brain structure and function. Thus, I find the general approach of the study sound and the obtained data valuable. The paper is well written and reasonably concise.

Meanwhile, there are several aspects of the paper that need clarification or improvement.

Major comments

1. One of the main concerns is the missing justification for the ordering of factors in the proposed causal chain: impulsivity → cortical thickness → risky behavior  (implied, e.g., by the following phrase in the abstract: "the identified cluster mediating the association between sensation seeking and risky behavior"). The authors should state clearly if this order was chosen a priori on theoretical grounds or obtained as the one that optimally corresponds to the measured data. If the first scenario is the case (suggested by the phrase "the theorized temporal ordering of the predictor (impulsive personality traits), ..."), it is important to provide theoretical arguments for this ordering. There are explanations in the Discussion as to why risky behavior is likely to be the last item in the chain, but not about the ordering of impulsivity and cortical thickness.

Additionally, as the data allow testing any order of causation, it would be valuable to know the variance explained by alternative models, such as:

a. cortical thickness → impulsivity → risky behavior

b. cortical thickness → impulsivity

    ↓

   risky behavior

the latter model implying a confounding rather than mediating role of cortical thickness for the other two factors. 

2. Another important issue is the insufficiently detailed description of statistical methods. The authors state: "To correct for multiple comparisons, we utilized a Monte Carlo simulation with 10,000 iterations and a cluster-wise threshold correction of p < .05."  Monte Carlo is a very broad class of algorithms, thus, the authors should specify with sufficient detail what simulation was performed, what assumptions about the data are required for its validity, and what methodological studies (if any) confirm its applicability and performance in the context of brain morphometry.

The same relates to the methods of mediation analysis: the types of variables (continuous, categorical etc.), distributional assumptions, regression models need to be described in the Methods section.

3. The use of self-report questionnaires is potentially affected by the variability in the attitude of the participants to the studied phenomena. Thus, in the case of RISQ, one person may consider a certain episode as impulsive eating (difficulty stopping eating), while another person may treat similar episodes as normal. Likewise, in UPPS-P, a very cautious person may answer 'no' to the item 'I am a cautious person' (thinking they are still 'not cautious enough'), while a less cautious person may answer 'yes'. It would be beneficial if the authors could discuss the potential for such subjectivity and its possible impact on the results.

Specific suggestions

(the numbers before the comments indicate the relevant line numbers in the manuscript)

2. "Cortical integrity", in the title and further in the text. I am not sure that the word "integrity" (meaning lack of damage) is applicable to a healthy population. It may be appropriate to use "thickness" instead (as is already done in most instances in the text).

152. "The thickness of the cortical mantle at each vertex was estimated using FreeSurfer’s [40] (v6) standard morphometric pipeline [41,42]." It is desirable to briefly describe the data processing pipeline (the method of normalization/nonlinear registration etc.).

184. In table 2, the type of studied correlation should be stated (Pearson, Spearman etc.). If Pearson, please indicate the reason for expecting a linear relationship between the variables.

It would be beneficial to provide a table with descriptive statistics (mean/SD) of all the measured parameters, including impulsive traits, risky behavior, and the BMI covariate. Additionally, it would be natural to include BMI in the correlation table (Table 2).

Another question regarding BMI is why it is included in the regression as a  covariate of no interest. Presumably, BMI may be influenced by impulsivity. Thus, regressing it out may remove part of the variability in cortical thickness related to impulsivity. Conversely, BMI was not included as a covariate in the mediation analyses. I suggest the authors provide rationale for these choices.

189. "...multiple regions in which thickness was associated with impulsive traits and cortical thickness"  The last mention of cortical thickness is probably a typo ("thickness associated with thickness").

254. Table 4 - please explain what are the numerical values in the rows starting with "Step 1(R^2)" and "Step 2(R^2)": .01, .07* etc. Apparently, they do not conform to the column format "beta(SE)" as they have no parentheses, and thus should be described separately.

302. "anterior cingulate gyrus" - this area is not mentioned in the Results, nor can it be seen in the figures 1 and 2. Please check this sentence and/or the results.

309. "The present finding of greater thickness in these regions extends research that has linked negative urgency to reduced grey matter volume in the dorsomedial frontal, temporal pole, and ventral striatal volumes in healthy adults [20]"  I suggest the authors discuss why these regions from the literature did not show a significant effect in the present study (e.g. a relationship with impulsive urgency, which includes negative urgency).

321. "greater thickness in the impulsive urgency superior parietal cluster was associated with greater frequency of past month risky behavior." According to Table 4, there is a significant association with lifetime risky behavior, not past month risky behavior. Please check this sentence and/or the results.

326. "whether sensation seeking precedes brain structure, or vice versa, is unclear"  This is related to the first major comment above. If the direction of causation is unclear, it would be logical to call the role of cortical thickness as either mediating or confounding the relation between impulsivity and risky behavior, without asserting that it must be mediation.

334. "clusters associated with sensation seeking and impulsive urgency were both independently related to recent engagement in risky behavior".
First, as mentioned above, according to Table 4, the cluster associated with impulsive urgency was related only with lifetime risky behavior, but not the recent (past month) risky behavior. Please check this sentence and/or the results.
Second, the word "independently" may be misleading here, and this is important for the interpretation of the findings. Indeed, it was known in advance that impulsive traits correlate with both risky behavior and cortical thickness in the pericalcarine cluster. Thus, to test the relationship between risky behavior and cortical thickness independently of impulsivity, an analysis controlling for impulsivity must be performed. The corresponding effect size is measured by partial correlation, and the significance can be tested using regression with impulsivity as a covariate of no interest. If there is significant effect in this analysis then the relationship between cortical thickness and risky behavior is not just a consequence of both parameters being correlated with impulsivity.

358. "whom reported" - should probably read "who reported"

358. "reported high rates of engagement in multiple types of risky behavior". Were these rates higher than the average for the whole population? If so, was this intended in the design of the study? And does this limit the scope of results to a high-risk (or high-impulsivity) subgroup? Please discuss these issues in the paper.

Reviewer 2 Report

Review of “Cortical Integrity Links Impulsive Personality Traits and Risky Behavior”

Manuscript ID brainsci-652203

The authors present a well-written manuscript that aims to identify the neuroanatomical underpinnings of dispositional risk taking and impulsive behavior by examining the structural correlates of impulsive traits. Specifically, they hypothesized that cortical thickness would vary as a function of the three-factor model of impulsive traits and that those differences in brain structure would partially explain the association between impulsive traits and engagement in risky behavior.

Images from 107 adults (after 8 exclusions) were used to test this hypothesis.

Table 1 provides sample characteristics. All characteristics are self-explanatory with the exception of “Justice System Involvement”. This is not mentioned elsewhere in the manuscript so some clarification regarding exactly what this means and/or how it was measured should be included. Given the topic of the study I presume that it indicates that those with justice system involvement have had some issues with law enforcement but the exact meaning of this is unclear as it could range from minor infractions such as receiving a parking ticket to major violations such as being convicted of a felony.

The explanations of the methods that were used as well as the data analytic strategy that was employed were clear and thorough.

In addition to finding several expected correlations among the self-reported factors, several clusters in the cortical mantle were identified that correlated with sensation seeking and impulsive urgency. However, conscientiousness was unrelated to cortical thickness. Using hierarchical linear regression, the authors found that thickness in these clusters was linked with recent and/or lifetime risky behaviors. A mediation analysis revealed that cortical integrity mediates the relationship between sensation seeking and recent risky behavior.

The mention of exploratory analysis in the discussion section (starting on line 285) is appreciated but it might be more appropriate to include this in the results section of the manuscript, with the discussion of this analysis remaining in the discussion section.

The lack of findings of any neuroanatomical correlates of conscientiousness is interesting and I welcome the authors to provide some speculation regarding why this might be in their discussion of the findings. It would also be worthwhile to include a suggestion for future research examining the predictive validity/utility of measurements of the clusters that were associated with sensation seeking and impulsive urgency. For example, future studies could measure these clusters and then provide participants with a laboratory risk-taking task to determine whether thickness in these clusters predicts risk-taking behavior. Alternatively, participants could be queried about their risk-taking behaviors during a period of time after the measurements were taken in order to establish whether thickness of the clusters that this work showed were related to past risky behaviors is predictive of future risky behaviors. Methods such as these would be of utility in establishing direction of causality (e.g., does the thickness of the clusters cause elevated risk taking behavior or does previous risk taking behavior result in increased thickness) in addition to paving a path for potential applications of this work. Although I strongly feel that the research presented in this manuscript represents an important initial step toward understanding the relationship between neuroanatomy and risk taking, the lack of causal directionality in the current work should also be mentioned as a limitation in the discussion section.

Overall, I feel that the current work would provide a valuable addition to the existing literature on the physiology underlying risk-taking. I am happy to recommend that it be accepted for publication following minor revisions that address the issues mentioned in my review. I would like to thank the authors for conducting this research and I hope that they will continue to conduct similar studies that aim to expand upon these findings in the future.

Round 2

Reviewer 1 Report

I thank the authors for addressing my comments. I have three additional remarks, the first being a comment on the present situation with method validation not requiring any specific manuscript editing, and the other two providing minor suggestions.

"This methodology has been validated for use with cortical thickness data [47]."
Without objecting to the inclusion of this sentence, I want to note for further consideration that, apparently, the cited study only considered false positive rates (FPRs) in a test comparing two groups of subjects, whereas the paper under review deals with a single-group test. This aspect has recently proved critical in a related (but distinct) context of cluster-wise thresholding in fMRI. Namely, in the paper [Eklund et al., 2016] discussed in [47] and in recent a follow-up study [Eklund et al., 2018], it was shown that between-group comparisons lead to nearly nominal FPRs under condervative cluster-forming thresholds, while a single-group test of the mean effect yields considerably inflated FPRs for most parametric thresholding approaches (in the fMRI setting). Thus, in future methodological studies of surface-based morphometry, it is highly desirable to extend the results of [47] to one-sample GLM tests for validating the FPR control in this case.

[Eklund et al., 2016] Eklund, A., Nichols, T.E., Knutsson, H., 2016. Cluster failure: why fMRI inferences for spatial extent have inflated false-positive rates. PNAS 113 (28), 7900–7905, doi: 10.1073/pnas.1602413113.
[Eklund et al., 2018] Eklund, Anders, Knutsson, Hans and Nichols, Thomas E. (2018) Cluster failure revisited: Impact of first level design and physiological noise on cluster false positive rates. Human Brain Mapping, 40 (7). pp. 2017-2032. doi: 10.1002/hbm.24350.

page 5
"All of the variables, except biological sex (men = 1; women = 0), were continuous and met the distributional assumptions for the mediation and regression models."

As far as I understand, the RISQ score is an integer (number of events) and is thus discrete, not continuous. I am not sure if this affects the results, but it is desirable to mention this.

page 6
"Vertex-wise analysis of the cortical mantle revealed multiple regions in which thickness was associated with impulsive traits and cortical thickness and survived correction for multiple comparisons (Figure 1, Table 3)."

Still there is "thickness associated with thickness". Please revise again.
